# The COVID-19 Pandemic and Its Impact on Homebound Nursing Students

**DOI:** 10.3390/ijerph17207383

**Published:** 2020-10-10

**Authors:** Juana Inés Gallego-Gómez, María Campillo-Cano, Aurora Carrión-Martínez, Serafín Balanza, María Teresa Rodríguez-González-Moro, Agustín Javier Simonelli-Muñoz, José Miguel Rivera-Caravaca

**Affiliations:** 1Faculty of Nursing, Catholic University of Murcia (UCAM), Guadalupe de Maciascoque, 30107 Murcia, Spain; jigallego@ucam.edu (J.I.G.-G.); mcampillo2@ucam.edu (M.C.-C.); sbalanza@ucam.edu (S.B.); mtrodriguez@ucam.edu (M.T.R.-G.-M.); 2Department of Cardiology, Instituto Murciano de Investigación Biosanitaria (IMIB-Arrixaca), Hospital Clínico Universitario Virgen de la Arrixaca, CIBERCV, 30120 Murcia, Spain; auroracm@hotmail.es (A.C.-M.); jmrivera429@gmail.com (J.M.R.-C.); 3Liverpool Centre for Cardiovascular Science, University of Liverpool, Liverpool L7 8TX, UK

**Keywords:** pandemic, COVID-19, nursing student, online exam, stress, physical exercise

## Abstract

University students are predisposed to stress, which could be even higher in Nursing students since they are directly related to the COVID-19 pandemic given their health training and area of knowledge. Our purpose was to assess the stress levels of Nursing students before and during lockdown due to the COVID-19 pandemic in Murcia (Spain), its influence on taking an online exam and how it was affected by physical exercise. This was an observational and prospective study including Nursing students from the second year of the Nursing Degree from 3 February to 23 April 2020. Three measurements were performed: one before lockdown and two during lockdown. Stress increased substantially during lockdown. Financial, family or emotional problems, as well as physical exercise, also increased. Indeed, at 40 days of lockdown, those students with financial, family or emotional problems, and less physical exercise reported significantly higher stress levels. In addition, those who passed the online exam had lower stress levels compared to those who failed. In conclusion, during lockdown, stress in Nursing students increased. This could be triggered by students’ personal problems, and physical exercise may be used as a way to reduce stress. Academic performance was better in those students reporting less stress.

## 1. Introduction

The SARS-CoV-2 infection, which causes coronavirus disease 2019 (COVID-19), has become an international and global concern as a public health emergency. Identified in late 2019, COVID-19 was quickly characterized as a global pandemic in March 2020 [1]. For the general population, it posed a challenge at sanitary, economic, politic, and logistic levels, also affecting the psychological sphere [2]. This situation is transforming all aspects of people’s lives [1].

The negative impact of the SARS-CoV-2 virus is spreading socially, among other things, because of the speed of transmission and the lack of preparedness for the prevention and treatment of the causative disease. Although some people may be resistant to the stress associated with SARS-CoV-19, for many others this is likely to contribute to the exacerbation of existing mental disorders and the emergence of new stress-related disorders [1].

Many countries have asked people who have potentially come into contact with the infection to be isolated in their homes or in a quarantine facility [3]. Subsequently, other countries, including Spain, declared a state of alarm due to a health crisis, forcing the entire population to be confined to their homes, except for essential workers (health workers, police, supermarket employees, delivery personnel, etc.). With regard to educational centers and specifically universities, the decision of switching teaching to an online format was taken in accordance with the Royal Decree 463/2020 of the 14th of March, which declared the state of alarm for the management of the health crisis situation caused by the COVID-19 pandemic [4].

In times of conflict, people need their community. When “social distancing” is the new norm, we use technology more broadly to keep ourselves informed and connected to family and friends. The COVID-19 pandemic has demonstrated how connected we are within our communities and around the world, and has highlighted the degree of interconnectedness of our institutions, including medical, public health, political, economic, and educational institutions [5].

Among the people who were quarantined, there was a high prevalence of distress symptoms and psychological problems. Studies reported general psychological symptoms, emotional disturbance, depression, anxiety, low mood, insomnia, symptoms of post-traumatic stress, irritability, anger, emotional exhaustion and stress. Low mood and irritability were notable for their high prevalence [3,6]. University students are no exception, and actually, they are predisposed to high levels of stress. Examinations, family, financial and emotional problems are all triggers for stress [7,8]. If we add the current pandemic, being confined to their homes and the change of classes and exams from traditional to an online format to this known situation, everything can become much more complicated. For example, in a recent study in first-year university students in France, the majority of the students indicated that their level of anxiety increased since the beginning of the lockdown. This was particularly high in those students who remained in their usual residence in comparison to those who relocated. In addition, students who did not relocate reported higher stress levels in the financial and personal health domains [9]. 

Nursing students are a very specific group of students; they were directly related to the pandemic given their health training and area of knowledge. In fact, despite that nurses are the largest group of health professionals in the Spanish National Health System, the collaboration of nursing students was evidenced by the need to sign special contracts called “sanitary aid”, given the severity of the pandemic. Although these contracts were only used with Nursing students from the last year, students from all courses were affected by the feeling of being needed in such a dramatic situation, despite their inexperience and incomplete training.

Spain was one of the countries most affected by SARS-CoV-2 infection in Europe, not only during the first wave but also during the second. Health professionals and nurses in particular are playing an important role in the control of the pandemic and the management of COVID-19 patients. Nursing students have seen the high workload that this profession requires, and that nurses are especially subjected to high levels of stress and anxiety. Since they will be the nurses of the future, adequate training and mental health are central. Therefore, the objective of this study was to identify the level of stress of Nursing students before and during lockdown due to the COVID-19 pandemic in Murcia (Spain), its influence on the performance of an online exam, and how it was affected by physical exercise.

## 2. Materials and Methods 

### 2.1. Design and Sample

This was an observational and prospective study including a group of students from the Faculty of Nursing of the Catholic University of Murcia (UCAM) (Murcia, Spain). On 3 February 2020, prior to the appearance of the COVID-19 and the declaration of the state of alarm in Spain, stress levels were recorded in this group of Nursing students from the second-year, as part of a research study that was designed with the aim to reduce stress in this particular population. However, after the approval of a new law [4], which prohibited the free movement of people starting on 14th March, the intervention could not be carried out.

Since initial data were already recorded, we decided to re-design the study by collecting data two times more after the lockdown. Thus, the second measurement was carried out on 24 March 2020, at 3.30 p.m., i.e., 10 days after lockdown. At 4 p.m. on the same day, the students took the exam for the subject of Clinical Nursing. The third and last measurement was taken on 23 April 2020, i.e., 40 days after lockdown. Therefore, three measurements in overall were carried out. The students accessed the questionnaires through a link published on the virtual campus of the Nursing Degree. The purpose of the study was clearly explained to them and they have to sign an online informed consent for participation. The study was approved by the Catholic University of Murcia Ethics Committee. The ethical standards laid down in the Declaration of Helsinki were followed. 

### 2.2. Study Variables

The Student Stress Inventory-Stress Manifestations (SSI-SM) questionnaire was used to assess (by quantifying) stress levels [10,11]. This questionnaire was previously validated in nursing students [7], and included 19 items using a 5-point Likert-type score (from 1 = not at all, to 5 = completely) related to emotional, physiological and behavioral areas. The maximum score was 95 points and higher scores indicated higher perceived stress. The scale, in turn, was composed by 4 factors. Factor 1 “Self-concept” on personality, included ten items (items 1, 3, 4, 5, 6, 7, 12, 13, 14 and 19), Factor 2 “Sociability” contained five items (items 10, 11, 13, 16 and 18), Factor 3 “Uncertainty” associated with digestive disorders [10,11], had six items (items 2, 3, 4, 7, 12 and 17), and Factor 4 “Somatization” felt by the students as a result of stressful situations and the perception of lack of university attendance, with five items (items 8, 9, 10, 14 and 19) (Appendix A).

Other variables analyzed were sex, age, physical exercise, and the presence of financial, family or emotional problems. This information was recorded during the first measurement in an anonymized electronic case report form (eCFR) completed by every student included. Sex was recorded as male/female/non-declared; age was recorded as numerical data; whereas physical exercise was recorded as regular (yes/no) and as hours of physical exercise per week. The presence of financial, family or emotional problems was recorded as yes/no. This information was left to the discretion of the students, who were informed to indicate “yes” if, in their opinion, they considered that there were sufficient financial, family or emotional obstacles to produce a negative impact on their lives in the medium-long term.

Academic performance was also analyzed. It was evaluated with the score obtained on the exam held on 24 March for the subject “Clinical Nursing”. The score was classified as pass (≥5) or fail (<5), with values ranging from 0 to 10.

### 2.3. Statistical Analysis

Categorical variables were expressed as frequencies and percentages. Continuous variables were tested for normality with the Kolmogorov–Smirnov test and presented as mean ± SD or median and inter-quartile range (IQR), as appropriate.

Spearman’s coefficient was used to evaluate the correlation between continuous variables. The McNemar’s test was used to compare changes in responses to questions within the different measurements (for categorical ones). The Mann–Whitney U test was used to compare continuous variables between groups. The Student’s *t*-test or the Wilcoxon test, as appropriate, were performed to assess the change of continuous variables during the different measurements.

A *p* < 0.05 was accepted as statistically significant. Statistical analyses were performed using the SPSS V.21.0 for Windows (SPSS) (IBM, Armonk, NY, USA).

## 3. Results

Of the initial 142 eligible students, four students refused to participate in the study. The three stress measurements were obtained in the remaining 138 students. Of them, 78.3% were female, with a median age of 20 years (IQR 19–23).

When we analyzed financial, family and emotional problems, we found that before lockdown, 54 (39.1%) students reported problems. However, during the lockdown this figure increased significantly. Thus, after 10 days of lockdown, 78 (56.5%) (*p* < 0.001) students reported problems, and after 40 days, 70 (50.7%) (*p* = 0.014) students reported these problems (Table 1). 

Similarly, when assessing the practice of physical exercise, we found that before lockdown, 81 (58.7%) students practiced physical exercise regularly, changing to 93 (67.4%) (*p* = 0.058) 10 days after lockdown, and 107 (77.5%) after 40 days (*p* < 0.001) (Table 1). Indeed, the median hours of physical exercise peer week were 2 (IQR 0–4) before lockdown, increasing to 3 (IQR 0–5) after 10 days of lockdown (*p* = 0.072), and to 4 (IQR 1–6) after 40 days of lockdown (*p* < 0.001).

Regarding stress, the median self-reported stress before lowdown according to the SSI-SM was 40 (IQR 30.8–48.3). This increased to 41 (IQR 33–51) after 10 days from lockdown (*p* = 0.001), and to 41 (IQR 34.8–49.0) after 40 days (*p* = 0.004) (Table 2). A comparison of the score in every stress factor among the measurements is shown in Table 2.

Focusing on the different factors composing the SSI-SM, female students had a worse self-concept (*p* = 0.013) and higher uncertainty (*p* = 0.005) than men before lockdown. Ten days after lockdown, these two factors remained, and there were also statistically significant differences with the overall score of the SSI-SM (*p* = 0.027). After 40 days of lockdown, females continued having the worst self-concept (*p* = 0.019) and the highest uncertainty (*p* = 0.005) (Table 3).

In addition, stress levels were higher in students reporting financial, family or emotional problems before lockdown. This was also confirmed in all the stress factors and the overall score of the SSI-SM in the two measurements during the lockdown (all *p* < 0.05) (Table 2). On the other hand, there were not significant differences in reported stress levels before lockdown between students with and without regular physical exercise. After 10 days of lockdown similar results were observed. However, 40 days after lockdown, we found that students with regular physical exercise had lower stress levels based on the overall score of the SSI-SM (39 [IQR 32–48] vs. 45 [IQR 38–56], *p* = 0.014) (Table 3).

Finally, a significant inverse correlation of the SSI-SM score was observed with all the stress factors (R = −0.455; R = −0.455; R = −0.470; R = −0.497; respectively, all with *p* < 0.001).

In terms of academic performance, the Clinical Nursing exam was passed by 97 (70.3%) students, with a mean score of 5.9 ± 1.9 points. Importantly, we observed that the students who passed the exam (performed the same day than the measurement of 10 days after lockdown) reported lower levels of stress in all factors (all with *p* < 0.001) and at the overall SSI-SM, than those who did not pass the exam (38 [IQR 32–46] vs. 53 [IQR 45–58], *p* < 0.001). Age was slightly but significantly correlated with the grades obtained (R = 0.181, *p* = 0.034).

## 4. Discussion

In the present study, we found that lockdown may increase stress levels and financial, family and emotional problems. However, stress and other social problems could be mitigated by implementing practices that helped with relaxation and to some extent, to escape, such as physical exercise.

Stress is a major issue for university students. Examinations, clinical practices, and personal problems, among other issues, have an influence on the increase in stress, with a negative impact on academic performance [12]. In addition, the declaration of a pandemic with the approval of a state of alarm associated with the lockdown make this even more complicated for students. Indeed, previous studies have already indicated that this situation is causing an increase in mental health problems such as stress, anxiety and depression in the general population [3,6,13]. There are even studies suggesting that it may have a large influence on the mental health of young people [2], recommending the development of effective interventions for this population [14]. In our study, when comparing the evolution of stress, we observed that before the declaration of the pandemic, students had lower levels of stress than during lockdown. However, in the two measurements performed during lockdown, there were no statistically significant differences with the level of self-reported stress remaining similar at 10 and 40 days. The level of stress was slightly higher at 10 days, perhaps coinciding with the scheduling of an examination, but these results followed the same direction as another study carried out during lockdown in the adult population, where stress increased as the days passed [15].

Recent research has shown that female Nursing students generally have significantly higher stress levels than male students [7]. In another study conducted in the areas most affected by the COVID-19 outbreak in China, it was found that the prevalence of post-traumatic stress symptoms was significantly higher in females compared to males [16]. In our study, before lockdown, females had lower self-concept and higher uncertainty than males, increasing as the days went by. The same was true for overall stress, which was significantly higher for female students at 10 days of lockdown, coinciding with the scheduled examination.

On the other hand, economic, family and sentimental problems were often found in university students [7]. In the present study, as the days of lockdown extended, these problems increased. In this sense, it was observed that students with problems were those who had the highest levels of stress. It is undeniable that the COVID-19 pandemic has a negative impact on the economy [17]. This results in a higher risk of job loss and therefore in a higher probability of financial and housing precarities. As the lockdown advanced, the personal situation of students under this condition might get worse, the students were more prone to suffer a worsening of their previous economic, family or sentimental problems, reporting higher stress [18,19]. Regarding physical activity, some authors claim that regular physical activity over a period of time could be an effective strategy that positively affects academic performance, although this is not entirely clear [20]. In the present study, a striking situation arose. We found that the number of students practicing physical exercise increased as the days of lockdown progressed. The number of hours of exercise per week also increased. In the last measurement, after 40 days of lockdown, we observed that, by associating physical exercise with stress, there was a considerable decrease in the level of stress in the most athletic students. However, this is not a particular situation of our students, since a previous study in Spain already reported an increase in physical activity in university students during the lockdown [21].

In terms of academic performance, the results obtained in the exam that was carried out using an online format after 10 days of lockdown were very positive. If we compare these results with others published in a population with similar characteristics, the improvements in the results are significant [7]. As mentioned above, Nursing studies in Spain must be compulsorily face-to-face and are regulated by Royal Decree 1393/2007 from 29 October, which establishes the organization of official university education [22]. This regulation specifies that the exams for the Nursing Degree must be taken in person at the University. However, in light of the new situation in which we find ourselves, for the first time, online classes and exams were authorized throughout Spain [4]. This legislative change allowed students to continue their studies and even, as shown here, may improve their academic performance.

Education, and particularly health-related education, has abruptly changed to an online-based version. The use of the online methodology has improved student learning outcomes. Advances in consumer-level educational technologies are viewed as very promising for improving student learning experiences in Nursing and Medicine. The fusion of different technologies and methodologies may allow teachers to focus on the strengths of each, while mitigating the limitations that arise from independent use of only one mode [23,24]. However, the necessity of e-learning methods also implies limitations, for example, for older students, those living in rural areas, with work and family responsibilities, or with limited electronic resources [25]. 

Therefore, the experience related to virtual training in Nursing, acquired during the pandemic, could change traditional teaching practices and provide new educational opportunities. However, it is necessary an effort by all educational agents to make up for the deficiencies in the training of students detected after the face-to-face closure of the University. It is key, to guarantee competence and confidence in this new environment, that the faculties re-evaluate their policies and study plans and permanently incorporate innovative and alternative learning modalities that involve students and turn them into proactive agents of the learning process [26].

### Limitations

There are some limitations regarding this study that should be noted. Due to the circumstances in which the study was initiated, the access to the initial sample was limited and could not be expanded, since the first measurement was aimed at an intervention study to reduce stress. 

In addition, we cannot guarantee a causal association between our observations, and therefore all results should be interpreted with caution and investigated further. Moreover, economic, family, or sentimental problems were assessed subjectively by the students. Then, it is possible that not all the students had the same consideration about what is or what is not a “problem”.

## 5. Conclusions

The need for quarantine and the severity of the COVID-19 pandemic has produced anxiety and stress in several people around the world. Nursing students are affected in relation to this pandemic, since they will be healthcare professionals on the front line in similar situations in the future. In addition, they had to quickly adapt to a new non-classroom teaching format. This combination of factors could trigger a high stress level in Nursing students. 

In the present study, we found that reported stress level increased significantly during lockdown in comparison to prior to lockdown. Similarly, the proportion of students reporting financial, family and emotional problems, as well as physical exercise, also increased. A relationship between family and emotional problems and higher stress levels was observed at 10 and 40 days of lockdown. On the contrary, at 40 days of lockdown, those students who practiced more physical exercise reported lower stress. Of note, in terms of academic performance, the results from the online learning format were positive. Most of the students passed the exam and students who passed had lower stress levels. 

In our opinion, further studies should investigate if stress levels decreased after completing the lockdown and if stress showed an increasing tendency during the second wave. It would also be interesting to explore novel approaches for reducing stress in this population and their results.

## Figures and Tables

**Table 1 ijerph-17-07383-t001:** Changes in the proportion of students reporting financial, family or emotional problems and physical exercise before and during the lockdown.

Analyzed Variables	Before Lockdown	10 Days after Lockdown			40 Days after Lockdown		
	N (%)	N (%)	* McNemar Test	* *p*-Value	N (%)	** McNemar Test	** *p*-Value
Financial, family or emotional problems	54 (39.1)	78 (56.5)	13.09	*p* < 0.001	70 (50.7)	6.73	*p* = 0.014
Physical exercise	81 (58.7)	93 (67.4)	4.23	*p* = 0.058	107 (77.5)	21.12	*p* < 0.001

* For ‘before lockdown’ vs. ‘10 days after lockdown’ comparison. ** For ‘before lockdown’ vs. ‘40 days after lockdown’ comparison.

**Table 2 ijerph-17-07383-t002:** Variation in the score of the stress factors and the overall SSI-SM score before lockdown, 10 days after lockdown, and 40 days after lockdown.

Factors of the SSI-SM	Before Lockdown Median (IQR)	10 after Lockdown Median (IQR)	40 after Lockdown Median (IQR)	1–2 Comparison	1–3 Comparison	2–3 Comparison
				* *p*-Value	** *p*-Value	*** *p*-Value
Factor 1 Self-concept	23 (18.0–30.0)	24 (19.0–30)	24.5 (19.0–30)	*p* = 0.013	*p* = 0.012	*p* = 0.944
Factor 2 Sociability	9 (6.0–13.0)	9 (7.0–13.0)	9 (6.8–12.0)	*p* = 0.021	*p* = 0.067	*p* = 0.636
Factor 3 Uncertainty	15 (12.0–18.0)	15 (12.0–19.0)	16 (12.0–19.0)	*p* = 0.048	*p* = 0.004	*p* = 0.468
Factor 4 Somatization	8 (6.0–10.0)	8 (6.0–11.0)	8 (6.0–10.0)	*p* = 0.001	*p* = 0.105	*p* = 0.146
Average SSI-SM score	40 (30.8–48.3)	41 (33.0–51.0)	41 (34.8–49.0)	*p* = 0.001	*p* = 0.004	*p* = 0.767

* For ‘before lockdown’ vs. ‘10 days after lockdown’ comparison. ** For ‘before lockdown’ vs. ‘40 days after lockdown’ comparison. *** For ‘10 days after lockdown’ vs. ‘40 days after lockdown’ comparison. SSI-SM = Student Stress Inventory-Stress Manifestations.

**Table 3 ijerph-17-07383-t003:** Relationship between the Student Stress Inventory-Stress Manifestations (SSI-SM) score and sex, financial, family or emotional problems, and physical exercise, before lockdown, 10 days after lockdown, and 40 days after lockdown.

Analyzed Variables	Factor 1 Self-Concept		Factor 2 Sociability		Factor 3 Uncertainty		Factor 4 Somatization		Average SSI-SM Score	
	Score Median (IQR)	*p*-Value	Score Median (IQR)	*p*-Value	Score Median (IQR)	*p*-Value	Score Median (IQR)	*p*-Value	Score Median (IQR)	*p*-Value
*Before lockdown*										
Male sex (*n* = 31)	21 (14–27)	0.013	8 (5–11)	0.264	12 (11–16)	0.005	7 (5–10)	0.239	37 (27–49)	0.058
Female sex (*n* = 107)	24 (19–30)		9 (6–13)		15 (13–18)		8 (6–10)		40 (33–48)	
Presence of financial, family or relationship problems (*n* = 54)	29 (21–31)	<0.001	11 (7–14)	0.002	16 (13–18)	0.018	9 (7–12)	<0.001	47 (34–53)	<0.001
Absence of financial, family or relationship problems (*n* = 84)	21 (17–27)		8 (6–10)		14 (11–16)		7 (5–9)		38 (28–43)	
Physical exercise (*n* = 81)	22 (17–30)	0.353	8 (6–11)	0.136	14 (11–17)	0.215	8 (6–10)	0.575	38 (29–48)	0.144
No physical exercise (*n* = 57)	24 (19–29)		10 (7–13)		15 (13–18)		8 (6–11)		40 (33–51)	
*10 days after lockdown*										
Male sex (*n* = 31)	21 (16–28)	0.012	8 (6–12)	0.100	13 (10–17)	0.003	7 (5–7)	0.067	37 (29–46)	0.027
Female sex (*n* = 107)	25 (20–31)		10 (7–13)		16 (13–19)		8 (7–11)		44 (36–52)	
Presence of financial, family or relationship problems (*n* = 78)	28 (19–31)	0.049	10 (7–13)	0.038	17 (12–20)	0.034	9 (7–12)	0.010	46 (34–64)	0.024
Absence of financial, family or relationship problems (*n* = 60)	24 (18–28)		9 (7–10)		14 (12–17)		8 (6–10)		39 (33–48)	
Physical exercise (*n* = 93)	24 (18–30)	0.666	9 (7–13)	0.685	15 (12–19)	0.398	8 (6–11)	0.607	40 (33–52)	0.553
No physical exercise (*n* = 45)	24 (21–30)		10 (7–12)		16 (13–19)		8 (7–12)		42 (36–51)	
*40 days after lockdown*										
Male sex (*n* = 31)	20 (15–29)	0.019	8 (6–13)	0.339	13 (10–18)	0.005	8 (5–10)	0.192	35 (27–52)	0.071
Female sex (*n* = 107)	25 (20–30)		9 (7–11)		16 (14–19)		8 (7–11)		41 (37–49)	
Presence of financial, family or relationship problems (*n* = 70)	28 (23–32)	<0.001	11 (8–14)	<0.001	17 (14–20)	<0.001	10 (7–11)	<0.001	46 (39–54)	<0.001
Absence of financial, family or relationship problems (*n* = 68)	21 (18–26)		8 (6–10)		14 (11–17)		7 (6–9)		38 (30–45)	
Physical exercise (*n* = 107)	24 (18–30)	0.101	9 (6–11)	0.029	15 (12–18)	0.020	8 (6–10)	0.008	39 (32–48)	0.014
No physical exercise (*n* = 31)	26 (22–31)		11 (8–14)		17 (14–21)		10 (7–12)		45 (38–56)	

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
