# Peer review of "The COVID-19 Pandemic and Its Impact on Homebound Nursing Students"

_ijerph, 2020, doi:10.3390/ijerph17207383_

Round 1
Reviewer 1 Report
- The abstract should be concise, and repetitive descriptions should be avoided as much as possible. It is strongly recommended that authors adopt the format of this journal template.
- The layout of the article is messy and does not strictly follow the format of the template.
- I suggest using lockdown to replace the word confinement, which is more in line with our usual expression.
- line 44:The citation format does not comply with the citation requirements of this publication
- Line79, line97 and line113 can add subheadings. eg: 2.1 Design and sample/2.2 Study variables/2.3 Statistical analysis
- Line 99 this article mentioned that there are 19 survey items, but items 20, 21 and 22 appear in line 104, line 105 and line 105 respectively
- The article should explain the sample with economic, family or emotional problems, that is, how to define whether the sample has economic, family or emotional problems. What are the economic, family and emotional problems?
In addition:
- At present, there are a lot of researches about the epidemic on college students' examination, family, economics, and emotional problems. So where is the research gap between this research and other papers? The author emphasizes that nursing students are a specific group, so how are they different from students in other majors?
- In Table 2, the average score of the students who failed the exam exceeded 95 points in 4 factors, which exceeded the upper limit mentioned in line 101.
- I don’t know if the total value in Table 2 is the average of factor1 to factor4? If so, please check the correctness of the data.
Author Response
The abstract should be concise, and repetitive descriptions should be avoided as much as possible. It is strongly recommended that authors adopt the format of this journal template.
>>> Thank you for your suggestion. We agree, and for this reason we have modified the abstract. We think it is now more concise and easy to read. According to he journal’s guidelines, it has now 200 words and contains a single paragraph following the style of structured abstracts (Background, Methods, Results; and Conclusion), but without headings.
The layout of the article is messy and does not strictly follow the format of the template.
>>> Thanks for this comment. We have carefully revised the format of the article and corrected when appropriate.
I suggest using lockdown to replace the word confinement, which is more in line with our usual expression.
>>> We agree with the reviewer and his/her suggestion has been addressed.
Line 44The citation format does not comply with the citation requirements of this publication.
>>> Sorry for this mistake. We have corrected it.
Line 79, line 97 and line 113 can add subheadings. eg: 2.1 Design and sample/2.2 Study variables/2.3 Statistical analysis
>>> Thanks for your suggestion. We have included subheadings now.
Line 99 this article mentioned that there are 19 survey items, but items 20, 21 and 22 appear in line 104, line 105 and line 105 respectively.
>>> We apologize if this was not clear in the previous version. The original SSI-SM questionnaire included 22 items, but we used here the version that we have previously validated with 19 items. This has been clarified and the questionnaire has been included in the Supplementary Material.
The article should explain the sample with economic, family or emotional problems, that is, how to define whether the sample has economic, family or emotional problems. What are the economic, family and emotional problems?
>>> We are sorry if this information was missing in the previous version. We have expanded and clarify this in the methods section.
In addition:
At present, there are a lot of researches about the epidemic on college students' examination, family, economics, and emotional problems. So where is the research gap between this research and other papers? The author emphasizes that nursing students are a specific group, so how are they different from students in other majors?
>>> Thank you very much for this interesting comment. The Introduction section has been clarified and the need for this study has been emphasized. We hope this could clarify the issues raised by the reviewer.
In Table 2, the average score of the students who failed the exam exceeded 95 points in 4 factors, which exceeded the upper limit mentioned in line 101.
>>> We have to apologize for this. This was a mistake and it has been corrected in this revised version. Thank you for your attention. Please, note that previous Table 2 is now Table 3.
I don’t know if the total value in Table 2 is the average of factor1 to factor4? If so, please check the correctness of the data.
>>> We are sorry if this was not clear. The total value shows the average score of the four factors. Table 2 has been re-designed and we think it is now clearer. Please, note that previous Table 2 is now Table 3.
Reviewer 2 Report
I have several recommendations which could help the authors to improve the manuscript:
- The Abstract should be revised. I suggest using the following scheme: overall relevance of the topic - specific relevance of the topic for Spain - major gaps in theory or practice that have existed - how the author approaches the solution of these problems in the paper - methods - findings - implications of the findings in future studies or policies. Very focused, one-two sentences per point. Check the sentences - what is "Descriptive, observational and prospective study"?
- Extend the explanation of the relevance of the topic in the Introduction (especially, regarding Spain). A reader should be efficiently introduced to the specific relevance of the problem in the area under study.
- How does this study contribute to the literature and how do the findings agree or disagree with other studies? Please address these issues in more detail in the Discussion.
- Conclusion is too brief, it should be expanded. Please summarize relevance, author's response to existing gaps, major findings, limitations, and implications
Author Response
I have several recommendations which could help the authors to improve the manuscript:
The Abstract should be revised. I suggest using the following scheme: overall relevance of the topic - specific relevance of the topic for Spain - major gaps in theory or practice that have existed - how the author approaches the solution of these problems in the paper - methods - findings - implications of the findings in future studies or policies. Very focused, one-two sentences per point. Check the sentences - what is "Descriptive, observational and prospective study"?
>>> We agree with the reviewer, and for this reason we have modified the abstract, including the design of the study. In our opinion, this revised version is now clearer. Unfortunately, we cannot use the scheme suggested by the reviewer since per journal’s guidelines, the abstract must have 200 words maximum in a single paragraph following the style of structured abstracts (Background, Methods, Results; and Conclusion), without headings. We hope the reviewer could understand this.
Extend the explanation of the relevance of the topic in the Introduction (especially, regarding Spain). A reader should be efficiently introduced to the specific relevance of the problem in the area under study.
>>> Thank for this comment. The Introduction section has been clarified and we have expanded the relevance of this topic in our particular context.
How does this study contribute to the literature and how do the findings agree or disagree with other studies? Please address these issues in more detail in the Discussion.
>>> Thank you for this suggestion. More information has been added to the Discussion section.
Conclusion is too brief, it should be expanded. Please summarize relevance, author's response to existing gaps, major findings, limitations, and implications
>>> Thank you for comment. Following your suggestion, we have modified the conclusion and expanded it.
Reviewer 3 Report
I enjoyed reading your article. I am providing a review that I hope will allow you to strengthen your article.
Abstract
- The last few sentences starting on line 32 with "Those who passed..." seemed to just repeat information that has already been stated in the abstract. I don't think the sentences are needed.
Format
- Use bolded subheading with X.1 or X.2 numbering. Examples
- 2.1 Design and sample - line 79
- line 97
- line 113
- line 225
Results
This is the section that you could improve and it would dramatically improve your paper. While your tables may seem self- explanatory to you, they aren't to your readers. Take some time to really explain each table. How did you find the numbers you are presenting? What does each column or row heading mean? For example, Table 1 - you have No and Yes on both the heading of the columns and rows. What does No, no versus No, yes mean?
You are reporting p values. Explain how you are finding the pvalues and what is the threshold for the pvalues to be significant or not significant and why.
How are you using the values in the tables to quantify or qualify stress? Stress is one of your main conclusions, how are you pulling that information from the data presented in the tables?
Again, I enjoyed your paper. A more detailed explanation of how you transformed your data into your conclusions will strengthen your paper.
Author Response
I enjoyed reading your article. I am providing a review that I hope will allow you to strengthen your article.
>>> Thank you very much for your overall positive opinion about our article.
Abstract
The last few sentences starting on line 32 with "Those who passed..." seemed to just repeat information that has already been stated in the abstract. I don't think the sentences are needed.
>>> Thank you for your suggestion. We have modified the abstract overall and we think this revised version in now clearer. Unfortunately, we cannot use the scheme suggested by the reviewer since per journal’s guidelines, the abstract must have 200 words maximum in a single paragraph following the style of structured abstracts (Background, Methods, Results; and Conclusion), without headings. We hope the reviewer could understand this.
Format
Use bolded subheading with X.1 or X.2 numbering. Examples
2.1 Design and sample - line 79
line 97
line 113
line 225
>>> Thanks for your suggestion. We have included subheadings now.
Results
This is the section that you could improve and it would dramatically improve your paper. While your tables may seem self- explanatory to you, they aren't to your readers. Take some time to really explain each table. How did you find the numbers you are presenting? What does each column or row heading mean? For example, Table 1 - you have No and Yes on both the heading of the columns and rows. What does No, no versus No, yes mean?
>>> Thank you for this suggestion. We are absolutely agreed with you. The Results section has been modified now, and even some tables have been re-designed. In our opinion, the information is presented now better and the results are shown in a more interesting form. We hope these changes would be adequate for the reviewer.
You are reporting p values. Explain how you are finding the p-values and what is the threshold for the p-values to be significant or not significant and why.
>>> Thanks for this comment. This information is already reported in the Statistical analysis subsection of Methods.
How are you using the values in the tables to quantify or qualify stress? Stress is one of your main conclusions, how are you pulling that information from the data presented in the tables?
>>> Thank you for your question. Overall, Tables have been modified and improved in this version. Also, the titles of the tables help now in their correct interpretation. Both, Table 2 and Table 3 show median scores of the stress factors and the SSI-SM. How these scores are translated into stress is explained in the Methods section. Briefly, the SSI-SM questionnaire included 19 items using a 5-point Likert-type score (from 1 = not at all, to 5 = completely) in 4 factors. The maximum score is 95 points and higher scores indicated higher perceived or self-reported stress.
Again, I enjoyed your paper. A more detailed explanation of how you transformed your data into your conclusions will strengthen your paper.
>>> Thank you for your comments, they are really appreciated. We hope this revised version would be better for you.
Reviewer 4 Report
This is a research paper that identifies the impact of different factors of stress, namely self-concept, sociability, uncertainty, and somatization, on nursing students during the COVID-19 pandemic in Spain. I see how the research topic can be valuable, adding to health behavioural research for this special time. However, the content related to the literature review, theoretical framework, results, and discussion may need to be reorganized for clarity. For example, the introduction section would be strengthened if authors could conduct a thorough literature review regarding the stress of college students during the COVID-19 pandemic. Also, the authors may want to clarify whether if financial, family or relationship problems were increased among nursing students in your results. Limitations of this study also should be properly addressed even it was a descriptive study.
Specific comments are as follows:
Introduction section: Overall, this section could be strengthened by providing more literature on the topic of stress among college students during COVID-19. For example, the following article may be helpful for your study:
Husky, M. M., Kovess-Masfety, V., & Swendsen, J. D. (2020). Stress and anxiety among university students in France during Covid-19 mandatory confinement. Comprehensive Psychiatry, 102, 152191
Also, providing your theoretical framework in this study would enhance your statements.
Line 45. It would be helpful for your readers to understand what specific levels related to the challenge of COVID-19 here. Please elaborate them out.
Line 65-69. When you listed so many psychological symptoms, you may want to properly cite the sources of literature.
Materials and Methods section: In general, the authors only describe information related to their outcome variable (SSI-SM) in this section. Information related to other important variables is all missing. For example, how did they measure, sex, age physical exercise etc. A simple description such as (yes/no) would be very helpful for your readers to understand your measurement.
Line 94. Please clarify whether if your participants signed a paper-and-pencil informed consent form.
Line 103. It would be helpful for your readers to understand your measurement tool by presenting sample item statements in SSI-SM instead of numbers. Your readers may have limited knowledge of those item numbers.
Line 104. When you mentioned “uncertainty associated with digestive disorders”, you may want to properly cite the source of references.
Result section
Table one and two can be reorganized and presented in a better way. The following article from an intervention study may offer some thoughts of reorganizing your tables:
- Kelly, J. A., Somlai, A. M., DiFranceisco, W. J., Otto-Salaj, L. L., McAuliffe, T. L., Hackl, K. L., Heckman, T. G., Holtgrave, D. R., & Rompa, D. (2000). Bridging the gap between the science and service of HIV prevention: transferring effective research-based HIV prevention interventions to community AIDS service providers. American journal of public health, 90(7), 1082–1088. https://doi.org/10.2105/ajph.90.7.1082
Line 155. You may want to present information related to age in your tables. Presenting your participants’ socio-demographic characteristics may improve your study results.
Discussion section:
Line 174-177. This study is a descriptive study; therefore, you may want to discuss and explain your results carefully. For example, you may want to use “may” to present your results such as “confinement may increase levels and financial, family and emotional problems”
Line 211-214. I am not sure if those problems increase during pandemic because your table 2 shows that most students indicating those problems were at the timing of the first measurement.
Line 222. More examples for practical implications would be helpful to improve current services targeted to college students in Spain.
Line 225. Many limitations should be addressed such as
1. No causal associations. 2. Question related to financial, family and relationship problems was very general. 3. Social desirability bias. Etc.
Author Response
This is a research paper that identifies the impact of different factors of stress, namely self-concept, sociability, uncertainty, and somatization, on nursing students during the COVID-19 pandemic in Spain. I see how the research topic can be valuable, adding to health behavioural research for this special time. However, the content related to the literature review, theoretical framework, results, and discussion may need to be reorganized for clarity. For example, the introduction section would be strengthened if authors could conduct a thorough literature review regarding the stress of college students during the COVID-19 pandemic. Also, the authors may want to clarify whether if financial, family or relationship problems were increased among nursing students in your results. Limitations of this study also should be properly addressed even it was a descriptive study.
>>> Thank you for your comments and suggestions. We hope this revised version would be better for you.
Specific comments are as follows:
Introduction section: Overall, this section could be strengthened by providing more literature on the topic of stress among college students during COVID-19. For example, the following article may be helpful for your study:
Husky, M. M., Kovess-Masfety, V., & Swendsen, J. D. (2020). Stress and anxiety among university students in France during Covid-19 mandatory confinement. Comprehensive Psychiatry, 102, 152191
Also, providing your theoretical framework in this study would enhance your statements.
>>> Thank you. We agree regarding these suggestions and therefore we have added some text in the introduction. In our opinion, it is now more structured and clear.
Line 45. It would be helpful for your readers to understand what specific levels related to the challenge of COVID-19 here. Please elaborate them out.
>>> We have expanded this information. Thank you.
Line 65-69. When you listed so many psychological symptoms, you may want to properly cite the sources of literature.
>>> We absolutely agree. However, there are already cited when the next sentence finished: “Low mood and irritability were notable for their high prevalence [3,6]”.
Materials and Methods section: In general, the authors only describe information related to their outcome variable (SSI-SM) in this section. Information related to other important variables is all missing. For example, how did they measure, sex, age physical exercise etc. A simple description such as (yes/no) would be very helpful for your readers to understand your measurement.
>>> We agree with you. This information has been included.
Line 94. Please clarify whether if your participants signed a paper-and-pencil informed consent form.
>>> We have clarified that our participants signed an online-based informed consent form.
Line 103. It would be helpful for your readers to understand your measurement tool by presenting sample item statements in SSI-SM instead of numbers. Your readers may have limited knowledge of those item numbers.
>>> We agree with the reviewer. For this reason, tables have been clarified and we have included as Supplementary Material a copy of the SSI-SM.
Line 104. When you mentioned “uncertainty associated with digestive disorders”, you may want to properly cite the source of references.
>>> We have included the appropriate references.
Result section
Table one and two can be reorganized and presented in a better way. The following article from an intervention study may offer some thoughts of reorganizing your tables:
Kelly, J. A., Somlai, A. M., DiFranceisco, W. J., Otto-Salaj, L. L., McAuliffe, T. L., Hackl, K. L., Heckman, T. G., Holtgrave, D. R., & Rompa, D. (2000). Bridging the gap between the science and service of HIV prevention: transferring effective research-based HIV prevention interventions to community AIDS service providers. American journal of public health, 90(7), 1082–1088. https://doi.org/10.2105/ajph.90.7.1082
>>> Thank you. All tables have been changed. We think they are now clearer and easy to read. Thanks for the suggestion.
Line 155. You may want to present information related to age in your tables. Presenting your participants’ socio-demographic characteristics may improve your study results.
>>> Thanks for this interesting suggestion. Age could be actually central to the assessment of stress. However, in our opinion, the impact of age in this particular study was minimal, since the median age was 20 years, with a narrow interquartile range (19-23). For the reviewer, we have tested the correlation of age with the SSI-SM and found no significant results in either of the measurements (data not shown). Given that in the current form none of the tables would benefit from including age and that this information would add little to the article, we have decided maintaining tables as are shown now in this revised version.
Discussion section:
Line 174-177. This study is a descriptive study; therefore, you may want to discuss and explain your results carefully. For example, you may want to use “may” to present your results such as “confinement may increase levels and financial, family and emotional problems”
>>> Thank you for this suggestion. More information has been added to the Discussion section and we think this version has been enhanced now.
Line 211-214. I am not sure if those problems increase during pandemic because your table 2 shows that most students indicating those problems were at the timing of the first measurement.
>>> Regarding this information, probably the best table to assess this is current Table 1. Compared to the proportion of students reporting financial, family or emotional problems before lockdown, this proportion increased after 10 days of lockdown (39.1% vs. 56.5%, p<0.001) and after 40 days of lockdown (39.1% vs. 50.7%, p=0.014). This has been clarified in Table 1 and in the text.
Line 222. More examples for practical implications would be helpful to improve current services targeted to college students in Spain.
>>> Thank you for this interesting suggestion. We agree and therefore we have included some text regarding this issue in the Discussion section.
Line 225. Many limitations should be addressed such as:
- No causal associations. 2. Question related to financial, family and relationship problems was very general. 3. Social desirability bias. Etc.
>>> We agree with you. This section has been improved and more limitations acknowledged.
Round 2
Reviewer 2 Report
The author addressed my recommendations and revised the manuscript accordingly
Reviewer 3 Report
Great job editing the paper
Reviewer 4 Report
The authors have addressed concerns and suggestions from reviewers accordingly. The current version of manuscript is mature and refined.